# Prediction of Putative Epitope Peptides against BaeR Associated with TCS Adaptation in *Acinetobacter baumannii* Using an In Silico Approach

**DOI:** 10.3390/medicina59020343

**Published:** 2023-02-11

**Authors:** A. S. Smiline Girija, Shoba Gunasekaran, Saman Habib, Mohammed Aljeldah, Basim R. Al Shammari, Ahmad A. Alshehri, Ameen S. S. Alwashmi, Safaa A. Turkistani, Abdulsalam Alawfi, Amer Alshengeti, Mohammed Garout, Sara Alwarthan, Roua A. Alsubki, Nouran M. Moustafa, Ali A. Rabaan

**Affiliations:** 1Department of Microbiology, Saveetha Dental College and Hospitals, Saveetha Institute of Medical and Technical Sciences [SIMATS], Saveetha University, P.H. Road, Chennai 600077, India; 2Department of Biotechnology, DG Vaishnav College, Chennai 600106, India; 3Department of Medical Education, King Edward Medical University, Lahore 54000, Pakistan; 4Department of Clinical Laboratory Sciences, College of Applied Medical Sciences, University of Hafr Al Batin, Hafr Al Batin 39831, Saudi Arabia; 5Department of Clinical Laboratory Sciences, College of Applied Medical Sciences, Najran University, Najran 61441, Saudi Arabia; 6Department of Medical Laboratories, College of Applied Medical Sciences, Qassim University, Buraydah 51452, Saudi Arabia; 7Department of Medical Laboratory Sciences, Fakeeh College for Medical Science, Jeddah 21134, Saudi Arabia; 8Department of Pediatrics, College of Medicine, Taibah University, Al-Madinah 41491, Saudi Arabia; 9Department of Infection Prevention and Control, Prince Mohammad Bin Abdulaziz Hospital, National Guard Health Affairs, Al-Madinah 41491, Saudi Arabia; 10Department of Community Medicine and Health Care for Pilgrims, Faculty of Medicine, Umm Al-Qura University, Makkah 21955, Saudi Arabia; 11Department of Internal Medicine, College of Medicine, Imam Abdulrahman Bin Faisal University, Dammam 34212, Saudi Arabia; 12Department of Clinical Laboratory Sciences, College of Applied Medical Sciences, King Saud University, Riyadh 11362, Saudi Arabia; 13Basic Medical Science Department, College of Medicine, Dar Al Uloom University, Riyadh 12922, Saudi Arabia; 14Medical Microbiology & Immunology Department, Faculty of Medicine, Ain Shams University, Cairo 1181, Egypt; 15Molecular Diagnostic Laboratory, Johns Hopkins Aramco Healthcare, Dhahran 31311, Saudi Arabia; 16College of Medicine, Alfaisal University, Riyadh 11533, Saudi Arabia; 17Department of Public Health and Nutrition, The University of Haripur, Haripur 22610, Pakistan

**Keywords:** nosocomial infections, bioinformatics, BaeR, epitope peptides, in silico

## Abstract

*Background and Objectives*: The BaeR protein is involved in the adaptation system of *A. baumannii* and is associated with virulence factors responsible for systemic infections in hospitalized patients. This study was conducted to characterize putative epitope peptides for the design of vaccines against BaeR protein, using an immune-informatic approach. *Materials and Methods*: FASTA sequences of BaeR from five different strains of *A. baumannii* were retrieved from the UNIPROT database and evaluated for their antigenicity, allergenicity and vaccine properties using BepiPred, Vaxijen, AlgPred, AntigenPro and SolPro. Their physio-chemical properties were assessed using the Expasy Protparam server. Immuno-dominant B-cell and T-cell epitope peptides were predicted using the IEDB database and MHC cluster server with a final assessment of their interactions with TLR-2. Results: A final selection of two peptide sequences (36aa and 22aa) was made from the 38 antigenic peptides. E1 was considered a soluble, non-allergenic antigen, and possessed negative GRAVY values, substantiating the hydrophilic nature of the proteins. Further analysis on the T-cell epitopes, class I immunogenicity and HLA allele frequencies yielded T-cell immuno-dominant peptides. The protein–peptide interactions of the TLR-2 receptor showed good similarity scores in terms of the high number of hydrogen bonds compared to other protein-peptide interactions. *Conclusions*: The two epitopes predicted from *BaeR* in the present investigation are promising vaccine candidates for targeting the TCS of *A. baumannii* in systemic and nosocomial infections. This study also demonstrates an alternative strategy to tackling and mitigating MDR strains of *A. baumannii* and provides a useful reference for the design and construction of novel vaccine candidates against this bacteria.

## 1. Introduction

*Acinetobacter baumannii* (*A. baumannii*) was declared a pathogen of priority by the World Health Organization (WHO), and it has also been included as a member of the ESKAPE group of pathogens due to its multi-drug resistance [1]. *A. baumannii* is a Gram-negative non-motile coccobacillus, and is considered the most potent pathogenic species, inducing various recalcitrant infections associated with the respiratory system, skin and wounds, urinary tract infections and septicemia. Recently, its role in oral infections has been decoded, predominantly in oral mucosa and periodontal-associated infections [2]. It is also known to cause serious complications, especially in hospitalized patients, due to its multi-drug resistance, with high mortality rates. In our earlier reports from India, we documented the ability of MDR strains of *A. baumannii* [3], along with its plasmid-encoded resistance patterns, to exhibit resistance against cephalosporins [4], carbapenems [5], tetracyclines [6] and trimethoprim–sulfamethoxazole [5]. In addition to antimicrobial resistance, various virulence factors have been reported to be associated with *A. baumannii*, demonstrating that it is a predominant nosocomial pathogen.

*A. baumannii* exhibits an impressive capability to adapt to harsh environmental niches and overcome variations, thus prevailing in the hospital atmosphere. Its greater chance of survival is mainly attributed to its sensible two-component system (TCS), which allows it to sense variations in its habitat and modify the regulation of its phenotypes [7]. The two-component system canonically comprises two proteins: the histidine kinase (HK) and the response regulator (RR) [8]. The TCS encompasses 20 different RRs, efficient proteins that regulate vital pathways and mechanisms involved in *A. baumannii*’s metabolism [9]. Amidst many operons, it is fascinating to note that the BaeR gene/protein is involved in the crucial efflux pumps in *A. baumannii*, especially the AdeABC RND, AdeIJK and MacAB-TolC efflux pumps [10,11]. 

Antimicrobial resistance is among the most serious health-related issues worldwide. Over the past few years, no new antibiotics have been introduced on the market to prevent infections caused by Gram-negative bacilli [12,13]. Since bacteria are ever-evolving and developing resistance against available antibiotics, a limited number of drugs are currently available to treat infections [14]. Initially, all beta-lactam antibiotics, including penicillin, cephalosporin, monobactams and carbapenems, worked well against Gram-negative bacilli, but these bacteria gradually developed resistance against most of them [15]. Antibiotics generally work by inhibiting cell metabolism and replication and interfering with the cell wall or protein synthesis [16]. Hence, bacteria tend to develop defense mechanisms to protect themselves against antibiotics. These mechanisms are developed as a result of mutations at the gene level [12,15].

BaeRS constitutes BaeS as the histidine kinase and BaeR as the RR. Its homology with *E. coli* TCSs and cross-talk with *A. baumannii* BaeR TCSs in regulating overlapping regulons have also been documented. Deletion studies on *BaeR* have shown a decrease in the tolerance of *A. baumannii* to tannic acid with the presence of sucrose, modifying the osmotic stresses [17,18]. Sequential studies on BaeR thus portray the adaptive role of *A. baumannii* in efflux-pump-mediated antibiotic resistance and its regulation of virulence patterns via targeting BaeR [8]. As prophylaxis is the best alternative strategy, the present investigation applied a novel immuno-informatic approach to predict potent vaccine candidates for BaeR with the application of bio-informatic tools and databases. The tools were selected in such a way to evaluate the epitope peptides from BaeR and to assess its affinity for inducing both humoral and cellular immune responses. To the best of our knowledge, there is no vaccine of *A. baumannii* using this approach that has been reported yet. The design and evaluation of antigenic and immunogenic peptides from the TCS-mediated cognate response regulator BaeR revealed promising immunogenic and non-allergenic peptides with potential as components of vaccines against *A. baumannii* to induce efficient humoral and cellular immune responses.

## 2. Materials and Methods

### 2.1. Selection of BaeR Protein for A. baumannii

Using the UNIPROT database, FASTA sequences of BaeR protein from clinical strains of *A. baumannii* (Strain AYE, SDF, NCGM237) were retrieved (https://www.uniprot.org/) (Accessed on 11 June 2022). 

### 2.2. Mapping of B-Cell Epitopes

The retrieved FASTA sequence of BaeR was used as input for the B-cell epitope predictions using the IEDB database server (http://tools.iedb.org/main/bcell/) (Accessed on 11 June 2022). Linear B-cell epitope prediction involves using a variety of methods to sequentially characterize antigenic epitopes using amino acid input sequences. 

### 2.3. Antigenicity Predictions

The VaxiJen v2.0 server (http://www.ddg-pharmfac.net/vaxijen/VaxiJen/VaxiJen.html) (Accessed on 15 June 2022) was used to detect the protective antigens and to categorize the antigens based on their physico-chemical properties using independent alignment predictions. In an evaluation of their performance, these were classified as a protective antigen or another antigen based on a predetermined cut-off and prediction likelihood with a precision ranging from 70 to 80 percent [16]. The ANTIGENpro server (http://scratch.proteomics.ics.uci.edu/) (Accessed on 25 June 2022) makes a sequence-based, alignment-free and pathogen-independent prediction of the antigenic peptide using the reactivity data obtained from a microarray study for five pathogens. SOLpro predicts the propensity of a protein to be soluble upon overexpression in *E. coli* using a two-stage SVM architecture based on multiple representations of the primary sequence. It predicts the solubility level of selected epitopes. It is because the insoluble proteins are a major obstacle for many experimental-based studies. SOLpro serves this purpose where a sequence-based prediction is applied to predict the propensity of a protein to be soluble on overexpression and to prioritize targets in large-scale proteomics projects and to identify mutations likely to increase the solubility of insoluble proteins.

### 2.4. Physico-Chemical Property Analysis of the Predicted Proteins

For the evaluation of the physico-chemical properties of the predicted peptides from BaeR, such as molecular weight, pI, extinction coefficient, composition of amino acid, composition of atoms, half-life (predicted), stability of amino acids, aliphatic index and GRAVY, the ProtParam server (https://web.expasy.org/protparam/) (accessed on 26 June 2022) was applied [6]. 

### 2.5. Allergenicity and Toxigenicity Predictions

The AlgPred server (https://webs.iiitd.edu.in/raghava/algpred2/) (accessed on 26 June 2022) was used to develop a systematic method for predicting allergenic proteins from BaeR. The program uses a variety of methods, including comparing known epitopes to any protein region, IgE mapping, MEME/MAST allergen and motif predictions, SVM module and BLAST allergen-representative peptide (ARP) search (2890 allergens). In addition, it includes a hybrid method for predicting the allergenic properties of promising peptides (SVMc + IgE epitope + ARPsBLAST + MAST). Furthermore, the epitopes’ toxigenic properties were determined using toxinpred, which evaluates the peptide input sequences based on the SVM score, along with the parameters of hydrophobicity, hydrophilicity and hydropathicity. 

### 2.6. Signal Location of the Epitope Peptides

The SignalP 4.1 server (https://services.healthtech.dtu.dk/service.php?SignalP-4.1) (accessed on 27 June 2022) was utilized to evaluate the sequence options based on the neural networks, as well as to validate the position and signal regions of BaeR peptides from *A. baumannii* [18,19]. 

### 2.7. Continuous Antibody Epitope Prediction

With the IEDB server, the location of the continuous epitopes, i.e., antigenic peptides from BaeR, was determined as an empirical rule based on parameters such as stability, hydrophilicity, exposed surface antigenic tendency, accessibility, polarity and turns. The antibody epitope predictions were evaluated based on six popular predictions [20]. In addition, the IEDB server will render an assessment on the correlation between the position of continuous epitopes and properties of polypeptide chains such as hydrophilicity, flexibility, accessibility, turns, exposed surface, polarity and antigenic propensity. The site of continuous epitopes may also be predicted from specific characteristics of the protein sequence. For predictions, a propensity scale was computed for each of the 20Aa using the score for a given residue I, a window size of ‘n’ and I—[n − 1]/2 neighboring residue. The X and Y axes represent residue positions and ratings, respectively, and the yellow portion of the output graphs represents the epitopes predicted based on the default threshold value. Bepipred parameters, such as the hydrophilicity, flexibility, accessibility, turns, exposed surface, polarity and antigenic propensity of polypeptides chains, were correlated with the location of continuous epitopes. This led to a search for empirical rules that would allow the position of continuous epitopes to be predicted from certain features of the protein sequence.

### 2.8. T-Cell MHC Class I and MHC Class II Epitope Predictions

Using the IEDB-AR server for consensus calculations based on CombLib, ANN and SMM, in addition to NetMHCpan-EL, the predicted antigenic epitopes from BaeR that could be recognized by T-cells were assessed for MHC class I binding. Consensus > ANN > SMM > NetMHCpan EL > CombLib were chosen in decreasing order against the set alleles by default in the database. A consensus approach was used to calculate binding potentials, which combined stabilization matrix alignment and the average relative binding matrix. The percentile ranks and the binding affinities under three separate categorizations based on IC50 values were evaluated to shortlist the predicted epitopes.

### 2.9. Class I Immunogenicity Predictions and Conservancy Analysis

In the prediction of immunogenicity, the ability of the immune-dominant peptides to evoke an immune response is the most critical criterion. This is accomplished by identifying epitopes that produce positive values using the IEDB server’s default parameters as potent immunogens. The IEDB conservancy analysis server renders the epitope conservancy as an additional measure to validate the epitopes. 

### 2.10. Cluster Analysis of the MHC-Restricted Alleles

The HLA alleles, which comprise a cluster of MHC alleles with selected peptides from BaeR, were deduced using the MHC cluster v2.0 server to determine functional relationships and evaluated based on the output obtained from the graphical tree and heat map (static) between the clusters. The functional relationship between the molecules with the MHC system is predicted by specific binding or clustering. 

### 2.11. Protein–TLR2 Receptor Interactions

The Galaxy web server was used to determine protein–peptide binding and observe potential interactions of the predicted epitopes with the TLR2 receptor. This step is crucial for effective vaccine design, as the optimization and structure of vaccines depend on interaction similarity scores and can be evaluated in terms of hydrogen bonds.

## 3. Results

### 3.1. BaeR B-Cell Epitope Prediction

Peptide mapping of the 5 FASTA sequences (228aa) of BaeR with the UNIPROT ID, V5VA19_ACIBA, B0V538_ACIBY, B0VRE0_ACIBS, A0A0Q1TBU4_ACIPI and A0A0E1PMP7_ACIBA (UNIPROT ID: Q8RMF4) from five different strains of A. baumannii on IEDB B-cell linear epitope prediction yielded 38 epitopes (default threshold set to >0.5) with the selection of two epitopes (E-1: position 65–100 and E-2: 163–184) for further analysis (Table 1). FASTA sequenced under B-cell linear epitope predictions yielded two common sequences designated as E1 and E2, which were taken for further analysis. The B-cell linear epitope predictions can be used to assess the input FASTA sequences and will yield possible epitopes with start and end regions. From the predicted epitopes, the selection of the final epitopes can be carried out upon further computational analysis. Figure 1 depicts yellow peaks based on the biochemical parameters, such as the composition of the amino acids, hydrophilicity, hydrophobicity, accessibility to the surface and flexibility.

### 3.2. Vaccine Properties

#### 3.2.1. Antigenic Potentials

ANTIGENpro server (http://scratch.proteomics.ics.uci.edu/) (Accessed on 25 June 2022) was used to analyze E1 as an antigen with a score of 0.260702 and E2 as a non-antigen with a score of 0.706722 (Table 2). E1 was determined to be a potential vaccine candidate, having the highest score of 0.4706, as predicted using the VaxiJen server 4.0 with a default threshold of >0.4. The epitopes analyzed under the server showed probable antigenic properties in comparison with the set default value. SolPro analysis showed both epitopes to be soluble, with E2 scoring the highest. 

#### 3.2.2. Allergenic and Toxigenic Properties

The prediction of the allergenic properties using the combined hybrid approach (SVMc / IgE epitope + ARPsBLAST + MAST) showed E1 was a non-allergen. SVM-AA predicted E2 as an allergen, while the other approaches showed it to be a non-allergen (Table 3). Toxinpred showed both epitopes to be non-toxins, with E1 showing an SVM score of −1.03 and a charge of −2.0 and E2 having an SVM score of −1.39 and a charge of −1.00. 

### 3.3. Physico-Chemical Analysis of the Peptides

The ProtParam assessments used to evaluate the physical and chemical parameters of E1 and E2 showed E2 as a stable protein with a shelf life of 3.5 h (in vitro). Both peptides had a good molecular weight with GRAVY values (negative) indicating hydrophilicity (Table 4). The aliphatic index value was high for E1 compared to E2. With an Aa/protein total liquid charge of 0, ProtParam was also used to analyze the iso-electric points, in comparison with the constant equilibrium points. This was high for E2 at a pH of 6.48. The molecular formulae of E1 and E2 were deduced as C_174_H_293_N_51_O_57_S_3_ and C_111_H_168_N_32_O_35_, respectively.

### 3.4. Signal Peptide Analysis

The location of epitopes E1 and E2 was predicted based on the transmembrane protein neural network using Signal P 4.1. This is a valid method of evaluating the epitope as a transmembrane protein, aiding the successful delivery of drugs in host cells. The tool has a set default value and transmembrane signals with a D cut-off value of 0.51. The TM neural network was not predicted for either epitope (Table 4, Figure 2). The signal P server yields three types of scores, viz., the C score, the value of the predicted cleavage site; the S score, the value of the predicted signal peptide; and the Y score, the sum of the C and S scores.

### 3.5. Selection of the Immune-Dominant T-Cell Epitopes

The varying lengths (9–10Aa) of immunogenic peptides were predicted using the percentile rank (≤0.2) as an indication of high eliciting affinity, where ANN/SMM IC50 values of less than 200 nm were predicted epitopes. The binding of MHC class I and class II alleles was evaluated using the IEDB-AR server based on the consensus combinatorial score (Appendix A). Interaction of the HLA alleles with the T-cell class I immuno-dominant peptides was assessed with the same IEDB server with the default settings. The interactions of the immuno-dominant peptides (n = 10) with T cells were also considered. The binding affinities, especially those associated with class I molecule restriction, were found to be 100-percent conserved, with six peptides scoring positively (Table 5).

### 3.6. MHC Restrictions and Cluster Analysis

Reassessment of the interaction of T-cell dominant peptides with frequent and predominant HLA alleles set as the default in cluster analysis of MHC revealed the graphical tree and the static heat maps (Appendix A). 

### 3.7. Protein–Peptide Interactions

The Galaxy web server was used to evaluate the similarity between the amino acids in the predicted epitopes (E1 and E2), which were aligned in such a way as to link the residues in the template TLR-2 to the template amino acids (Figure 3). E1 had the most peptide interactions, with a maximum number of 18 hydrogen bonds and an interaction similarity score of −27 Kcal/mol. E2 showed 14 hydrogen bonds, with an interaction similarity score of −19 Kcal/mol. Figure 3 shows the scaled stick-and-ball model for the obtained interactions. 

## 4. Discussion

In the present study, we assessed potent epitope peptides that could be selected as vaccine candidates to target the BaeR protein associated with the TCS of *A. baumannii*. This research work was conducted as an observational computational/in silico analysis to identify immuno-dominant peptides targeting BaeR in *A. baumannii* by employing a computational bio-informatics approach. A crucial step in this approach is the selection of B-cell and T-cell epitopes that could elicit an immune response in the host. The high availability of proteomic and genomic resources in a common computational in silico platform paves the way for this approach as a cost-effective measure for the prediction and selection of novel *BaeR* vaccine candidates. Many such studies have been conducted using an immune-informatic approach [19,20], and a similar study was carried out to evaluate the vaccine epitope from the GacS sensor system in *A. baumannii* [21].

BaeR was selected amongst many TCSs, due to its vital role in enhancing the virulence in *A. baumannii*, enabling it to survive under different environmental niches. The UNIPROT database yielded a total of nine BaeR proteins, including the proteins from other *Acinetobacter* sp., (viz., *A. pitti*, *A. calcoaceticus*, *A. hemolyticus*), with five BaeR proteins submitted from the clinical strains of *A. baumannii*, which were selected for further analysis, considering the severity of their pathogenicity and virulence. B-cell linear epitope predictions for the retrieved FASTA sequences from *BaeR* showed only two common sequences among the total predicted epitopes. Thus, we designated the two selected common epitopes from positions 65–100 and 163–184 as E1 and E2 for further analysis [22,23]. The IEDB B-cell epitope tool is useful for predicting epitopes based on various parameters with set default scores, allowing for the evaluation of antigenic regions’ capability to elicit the humoral immune response. In Figure 1, the yellow peaks indicate antigenic regions, which were computed with ease using prediction scores with high threshold values.

The ANTIGENPro, SolPro and VaxiJen servers produced promising antigenic predictions for E1 and E2, with the set default parameters and by comparison of the threshold values [21,22]. The probable antigenicity was predicted based on the set default value of >0.4, and E1 was considered an antigen and both epitopes were determined to be soluble under the SolPro server. SOLpro predicts the likelihood a protein will be soluble upon overexpression in *E. coli* using a two-stage SVM architecture based on multiple representations of the primary sequence. It predicts the solubility level of selected epitopes. The solubility of both peptides was noted, and peptide E2 was determined to be a non-antigen despite having the highest score for antigenicity. E2 was partially an allergen, according to its SVM module as evaluated using AlgPred, though the other approaches indicated E2 as a non-allergen. Potent vaccine candidates were selected based on the ‘z’ score by the VaxiJen server, which uses the physico-chemical parameters of the submitted peptides devoid of sequence alignment and ANTIGENPro through the default microarray data of pathogens in the server [23,24]. In the Algpred approach, the server will search known IgE epitopes in query protein sequence and will assign as allergen if any segment have high similarity with any known epitope. If there is a known epitope(s), then mapping of the epitope(s) is performed in the query sequence. The specificity of this approach is very high, but the disadvantage is that it has low sensitivity, as not all IgE epitopes of all allergens are known. E1 was determined to be a non-allergen based on the SVM and hybrid approach at a set default of 0.4. 

Analysis of the physio-chemical properties of E1 and E2 was easily achieved using the Protparam server, which is a useful tool in experimental in vitro studies [24]. E2, being considered a non-antigen, was deduced to be a stable protein with a high shelf life of 3.5 h when compared to E1, which was interpreted with the three model organisms yeast, human and *E. coli*. However, E1 possessed the highest molecular weight, aliphatic index and GRAVY values, and a shelf life of just 1 h. The negative GRAVY values of the epitopes indicated they had good hydrophilicity, as evaluated with the set formula in the server and ratio of the sum of the Aa hydropathy values to the number of residues in the sequences. Based on this, both E1 and E2 scored as hydrophilic peptides. In addition to the results in Table 4, interpretations about the number of atoms, extinction coefficients, N-terminal details, etc., can also be assessed using the server [23].

We also assessed the signal localization of E1 and E2 as this is yet another crucial step in the optimization of drug delivery via minimizing the time and cost. With the advent of bioinformatics, this can be achieved using tools such as SignalP servers to develop a further understanding of the drugs and vaccine development. In the context of inducing an efficient immune response, the HLA allele restriction of T-cell interaction with the selected epitopes was also computationally assessed in the present study. We selected the first five peptides that were predicted to act against MHC class I and II alleles due to having the lowest percentile ranks together with IC_50_ values (<200 nm) under ANN, SMM, CombLib and NetPan scoring and ranking [20]. 

In this study, the functional relationships between interactions were successfully assessed using heat maps and graphical trees of epitopes with common HLA alleles. The MHC cluster server was used for the algorithmic annotations, with red and yellow suggesting strong and weak interactions, respectively. The interaction similarity scores were used to interpret the protein–peptide complexes, which were indicated as hydrogen bonds devoid of any binding energy calculations. These interactions show a strong binding between the predicted epitopes with the HLA alleles and its divergent patterns. Because of the small number of amino acid sequences, modeling of the epitope protein was not possible. 

## 5. Conclusions

The present study is the first of its kind and involved a novel attempt to predict a few vaccine epitopes from the BaeR of *A. baumannii* to tackle and mitigate MDR strains of *A. baumanii* via computational approaches. However, the deduced epitopes require further experimentation and validation in animal models, and the limitations of their immune response based on the physical condition of the host must be analyzed. We conclude by highlighting the possibility of evaluating vaccine epitopes for *A. baumannii* with the diligent application of subtractive genomics and reverse vaccinology technology.

## Figures and Tables

**Figure 1 medicina-59-00343-f001:**
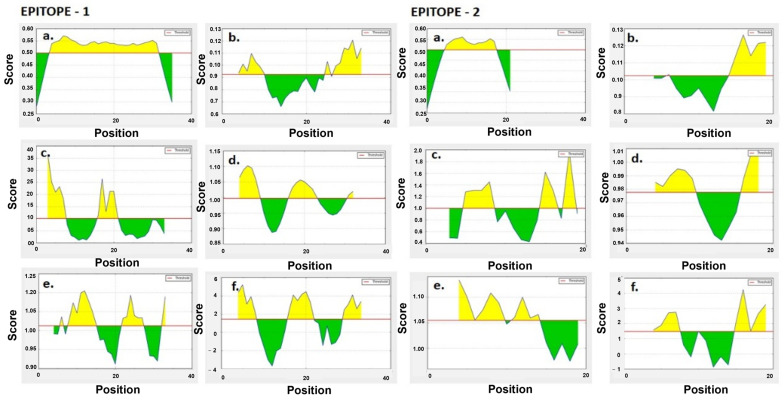
Antigenic predictions of B-cell epitope of BaeR from *A. baumannii* (epitope 1 and epitope 2); start and end positions of yellow peaks show the sequence of antigenic peptide as determined by (**a**) BepiPred linear epitope predictions, (**b**) Chou–Fasman beta turn assessment, (**c**) Emini surface accessibility predictions, (**d**) Karplus–Schulz flexibility predictions, (**e**) Kolaskar–Tongaonkar antigenicity and (**f**) Parker hydrophilicity assessments.

**Figure 2 medicina-59-00343-f002:**
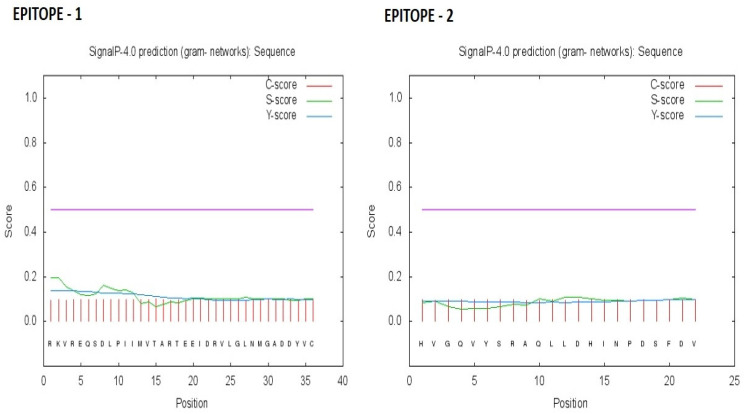
Signal P—no TM neural predictions (D cut-off value) using Signal P 4.0 server (C score—pink, S score—green and Y—blue, Purple—cut off value).

**Figure 3 medicina-59-00343-f003:**
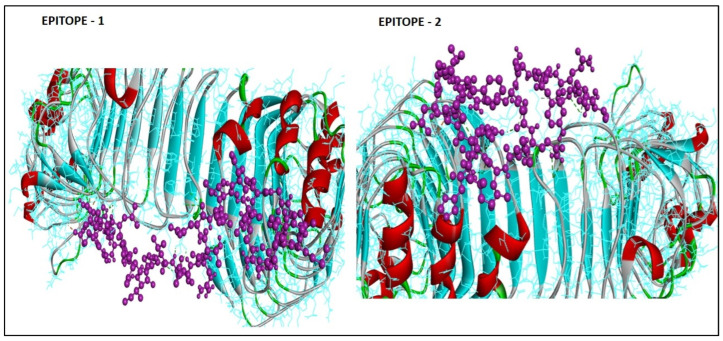
Protein–peptide interaction of the predicted BaeR epitopes with TLR-2 receptors (scaled ball model of the epitope peptide—pink and line model of the TLR2 receptor—blue).

**Table 1 medicina-59-00343-t001:** Predictions of B-cell linear epitopes from BaeR of *A. baumannii*.

Protein ID	Strain	Start	End	Predicted Peptides	Length
V5VA19_ACIBA	-	5	13	MLVEDEVEL	9
		26	100	FEVSMFHDGQDAYTSFQQRKPNLMILDLMVPRMDGLTICRKVREQSDLPIIMVTARTEEIDRVLGLNMGADDYVC	75
		125	133	PEQNDSFRI	9
		137	152	QQRIWYQQKSLSLTPT	16
		163	184	HVGQVYSRAQLLDHINPDSFDV	22
		201	214	TEVAETGNRHEWIQ	14
B0V538_ACIBY	AYE	5	13	MLVEDEVEL	9
		26	63	FEVSMFHDGQDAYTNFQQRKPNLMILDLMVPRMDGLTI	38
		65	100	RKVREQSDLPIIMVTARTEEIDRVLGLNMGADDYVC	36
		125	133	PEQNDSFRI	9
		137	152	QQRIWYQQKSLSLTPT	16
		163	184	HVGQVYSRAQLLDHINPDSFDV	22
		201	214	TEVAETGNRHEWIQ	14
		224	224	E	1
B0VRE0_ACIBS	SDF	5	16	MLVEDEVELAHL	12
		18	20	CDY	3
		23	23	A	1
		26	100	FEVSMFHDGQDAYTSFQQRKPNLMILDLMVPRMDGLTICRKVREQSDLPIIMVTARTEEIDRVLGLNMGADDYVC	75
		125	133	PEQNDSFRI	9
		137	152	QQRIWYQQKSLSLTPT	16
		162	184	EHVGQVYSRAQLLDHINPDSFDV	23
		201	214	TEVAETGNRHEWIQ	14
		223	224	FE	2
A0A090B602_ACIBA	-	5	13	MLVEDEVEL	9
		26	63	FEVSMFHDGQDAYTNFQQRKPNLMILDLMVPRMDGLTI	38
		65	100	RKVREQSDLPIIMVTARTEEIDRVLGLNMGADDYVC	36
		125	133	PEQNDSFRI	9
		137	152	QQRIWYQQKSLSLTPT	16
		163	184	HVGQVYSRAQLLDHINPDSFDV	22
		201	214	TEVAETGNRHEWIQ	14
		224	224	E	1
A0A0E1PMP7_ACIBA	NCGM 237	5	13	MLVEDEVEL	9
		26	100	FEVSMFHDGQDAYTSFQQRKPNLMILDLMVPRMDGLTICRKVREQSDLPIIMVTARTEEIDRVLGLNMGADDYVC	75
		125	133	PEQNDSFRI	9
		137	152	QQRIWYQQKSLSLTPT	16
		163	184	HVGQVYSRAQLLDHINPDSFDV	22
		201	214	TEVAETGNRHEWIQ	14
		223	224	FE	2

**Table 2 medicina-59-00343-t002:** Antigenicity and solubility analysis of the predicted BaeR epitopes using VaxiJen, ANTIGENPro and SOLPro.

Peptide	Epitope	Peptide Sequence	VaxiJen		Antigen PRO	SolPro	
			Threshold Value (≥0.4)			Threshold Value (≥0.5)	
1	E1	RKVREQSDLPIIMVTAR TEEIDRVLGLNMGADDYVC	0.4706	Antigen	0.260702	0.897659	Soluble
2	E2	HVGQVYSRAQLLDHINPDSFDV	0.2225	Non-Antigen	0.706722	0.931871	Soluble

**Table 3 medicina-59-00343-t003:** Allergenicity predictions for the BaeR epitopes (SVM and hybrid approaches at threshold value 0.4) using AlgPred.

Peptide	Predicted Antigens	IgE	MAST	SVM-Aa	SVM-dp	BLAST–ARP	Hybrid
E1	RKVREQSDLPIIMVTARTEEIDRVLGLNMGADDYVC	NA	NA	NA	NA	NA	NA
E2	HVGQVYSRAQLLDHINPDSFDV	NA	NA	A	NA	NA	NA

NA—non-allergen, A—allergen.

**Table 4 medicina-59-00343-t004:** Physio-chemical properties and SignalP 4.0 predictions for BaeR epitopes.

Peptide	Predicted Antigens	MW	IP	SI	SL	AI	GRAVY
E1	RKVREQSDLPIIMVTARTEEI DRVLGLNMGADDYVC	4107.73	4.66	40.86 (unstable)	1 h	102.78	−0.214
E2	HVGQVYSRAQLLDHINPDSFDV	2510.75	5.13	37.19 (stable)	3.5 h	97.27	−0.341
**SignalP 4.0 predictions for the BaeR transmembrane peptides**							
**Peptide**	**Predicted antigens**	**Score**			**Signal TM**		
E1	RKVREQSDLPIIMVTARTEEI DRVLGLNMGADDYVC	0.165			No		
E2	HVGQVYSRAQLLDHINPDSFDV	0.092			No		

MW: molecular weight, IP: iso-electric point, SI: stability index, SL: shelf life, AI: aliphatic index and GRAVY: grand average hydropathicity. Note: D cut-off signal—TM networks: 0.51; D cut-off signal—no TM networks: 0.57.

**Table 5 medicina-59-00343-t005:** Class I immunogenicity predictions of BaeR epitopes with positive values, as an indication of their ability to elicit an immune response and degree of conservancy.

Predicted Epitopes	T-cell Epitopes Predicted	Peptide Length	Score	Degree of Conservancy
E1: RKVREQSDLPIIMVTARTEEIDRVLGLNMGADDYVC	TEEIDRVLGL	10	0.28772	100%
	MVTARTEEI	9	0.28191	100%
	EEIDRVLGL	9	0.16522	100%
	LPIIMVTAR	9	0.1006	100%
	REQSDLPII	9	−0.12451	100%
E2: HVGQVYSRAQLLDHINPDSFDV	LLDHINPDSF	10	0.07467	100%
	LLDHINPDSF	10	0.07467	100%
	VYSRAQLLDH	10	−0.08316	100%
	VYSRAQLLD	9	−0.0884	100%
	GQVYSRAQL	9	−0.13736	100%

Note: positive values indicate increased immune response.

## Data Availability

More data related to this study can be accessed upon a reasonable request to the corresponding authors.

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
