# Peer review of "Prediction of Putative Epitope Peptides against BaeR Associated with TCS Adaptation in Acinetobacter baumannii Using an In Silico Approach"

_medicina, 2023, doi:10.3390/medicina59020343_

Round 1
Reviewer 1 Report
1. The majority of references are cited in the Intro which needs to be reduced. On the other hand, references in discussion need to increase. Some references are too old (<2012), and need to be removed or replaced.
2. Experimental verification is needed. At least one experimental data should be conducted to verify the prediction.
Author Response
Reviewer 1
Comments and Suggestions for Authors
- The majority of references are cited in the Intro which needs to be reduced. On the other hand, references in discussion need to increase. Some references are too old (<2012), and need to be removed or replaced.
Response: Dear reviewer, thank you for your valuable suggestion. We have revised the citations in the modified version of manuscript.
- Experimental verification is needed. At least one experimental data should be conducted to verify the prediction.
Response: Dear reviewer, we would like to appreciate that you have given us your valuable suggestion to perform at least one wet lab analysis on order to validate the vaccine construct. However, because of the less availability of lab facilities and also because of the budget issues, we are not able to run the wet lab analysis this time. This was one of the reasons also that we planned this insilico study, as we don’t have any funding to carryout the wet lab experiments. But we already applied for fundings in order to work on the invitro validation of these construct in near future. We have followed the following studies:
Naveed, M.; Ali, U.; Karobari, M.I.; Ahmed, N.; Mohamed, R.N.; Abullais, S.S.; Kader, M.A.; Marya, A.; Messina, P.; Scardina, G.A. A Vaccine Construction against COVID-19-Associated Mucormycosis Contrived with Immunoinformatics-Based Scavenging of Potential Mucoralean Epitopes. Vaccines 2022, 10, 664. https://doi.org/10.3390/vaccines10050664.
Naveed, M.; Yaseen, A.R.; Khalid, H.; Ali, U.; Rabaan, A.A.; Garout, M.; Halwani, M.A.; Mutair, A.A.; Alhumaid, S.; Al Alawi, Z.; et al. Execution and Design of an Anti HPIV-1 Vaccine with Multiple Epitopes Triggering Innate and Adaptive Immune Responses: An Immunoinformatic Approach. Vaccines 2022, 10, 869. https://doi.org/10.3390/vaccines10060869.
Naveed, M.; Jawad-ul-Hassan; Ahmad, M.; Naeem, N.; Mughal, M.S.; Rabaan, A.A.; Aljeldah, M.; Shammari, B.R.A.; Alissa, M.; Sabour, A.A.; et al. Designing mRNA- and Peptide- Based Vaccine Construct against Emerging Multidrug-Resistant Citrobacter freundii: A Computational-Based Subtractive Proteomics Approach. Medicina 2022, 58, 1356. https://doi.org/10.3390/medicina58101356.
Naveed, M.; Jabeen, K.; Naz, R.; Mughal, M.S.; Rabaan, A.A.; Bakhrebah, M.A.; Alhoshani, F.M.; Aljeldah, M.; Shammari, B.R.A.; Alissa, M.; et al. Regulation of Host Immune Response against Enterobacter cloacae Proteins via Computational mRNA Vaccine Design through Transcriptional Modification. Microorganisms 2022, 10, 1621. https://doi.org/10.3390/microorganisms10081621.
Reviewer 2 Report
Reasonable detail is provided regarding the biological role and mechanism of BaeR; however, the manuscript lacks sufficient rationale for the computational tools and databases used throughout the study. There is little indication of why this bioinformatic approach is an efficient route.
Moreover, the authors do not provide the means to evaluate the validity of this approach for identifying vaccine epitopes. The authors should consider using previously well characterized proteins with known vaccine epitopes to assess how appropriate this workflow actually is.
As an issue of reproducibility, the specific method (among multiple options and parameters) used within the IEDB B cell epitope prediction web server is not indicated by the authors. Depending on which of the BaeR strains (e.g. AYE, SDF, NCGM237) and the choice among multiple epitope prediction options, the results from the webserver can be drastically different. Using the author’s descriptions within Materials and Methods and Results, the reader is unable to recreate the results shown in Table 1.
The ‘Discussion’ fails to comment on the implications of their results. The authors should interpret their findings more deeply rather than simply stating what each of the webservers generated.
Figure quality is poor, making it difficult to evaluate the data and assess the validity of the authors conclusions.
The overall writing quality is poor (e.g. grammar and word choice) and in need of extensive revisions to improve readability. For example, grammatical mistakes need to be corrected (e.g. line 78, “TCS’s” should be “TCSs” since it is the plural and does not show possession). Articles and punctuation are incorrectly used in numerous places throughout the manuscript. (e.g. line 100, “These mechanisms are developed as result of mutations at gene level” should be “These mechanisms are developed as a result of mutations at the gene level.”)
The language used in the abstract could use greater precision or quantification. For example, how should the reader interpret “TLR-2 receptor showed good interaction similarity scores and high number of hydrogen bonds.” These are relative statements that need a reference point (e.g. good interaction similarity scores compared to what? High number of H-bonds compared to other protein-peptide interactions in this study or compared to experimentally validated results from the literature?).
Author Response
Reviewer 2
Comments and Suggestions for Authors
Reasonable detail is provided regarding the biological role and mechanism of BaeR; however, the manuscript lacks sufficient rationale for the computational tools and databases used throughout the study. There is little indication of why this bioinformatic approach is an efficient route.
Response: Dear reviewer, the rationale was included in the last paragraph under Introduction. Furthermore, a description about BaeR has been written and highlighted in yellow colour.
Moreover, the authors do not provide the means to evaluate the validity of this approach for identifying vaccine epitopes. The authors should consider using previously well characterized proteins with known vaccine epitopes to assess how appropriate this workflow actually is.
Response: Dear reviewer, thank you for your valuable suggestion. We have revised the statements as per your comments in the discussion section of revised manuscript. At line 317-320, similar such study is included in the first paragraph under discussion.
As an issue of reproducibility, the specific method (among multiple options and parameters) used within the IEDB B cell epitope prediction web server is not indicated by the authors. Depending on which of the BaeR strains (e.g. AYE, SDF, NCGM237) and the choice among multiple epitope prediction options, the results from the webserver can be drastically different. Using the author’s descriptions within Materials and Methods and Results, the reader is unable to recreate the results shown in Table 1.
Response: Line 211-216: Dear reviewer, thank you for highlighting the point here. The explanation has been given under the results section 3.1.
The ‘Discussion’ fails to comment on the implications of their results. The authors should interpret their findings more deeply rather than simply stating what each of the webservers generated.
Response: Line 316-319, 337-346, 350-351. Dear reviewer, we have revised the discussion section and added more implications on the results obtained.
Figure quality is poor, making it difficult to evaluate the data and assess the validity of the authors conclusions.
Response: Dear reviewer, we apologise for the bad quality figures in last version. The figure quality has improvised in the revised version of manuscript.
The overall writing quality is poor (e.g. grammar and word choice) and in need of extensive revisions to improve readability. For example, grammatical mistakes need to be corrected (e.g. line 78, “TCS’s” should be “TCSs” since it is the plural and does not show possession). Articles and punctuation are incorrectly used in numerous places throughout the manuscript. (e.g. line 100, “These mechanisms are developed as result of mutations at gene level” should be “These mechanisms are developed as a result of mutations at the gene level.”)
Response: Dear reviewer, we would like to appreciate that you have raised the issue of grammatical mistakes. The manuscript has been thoroughly revised for English proofreading and grammatical mistakes. To make the English content excellent, we have utilised the services from English experts to check the manuscript for grammatical mistakes. Furthermore, at line 108-109, we have revised the sentence.
The language used in the abstract could use greater precision or quantification. For example, how should the reader interpret “TLR-2 receptor showed good interaction similarity scores and high number of hydrogen bonds.” These are relative statements that need a reference point (e.g. good interaction similarity scores compared to what? High number of H-bonds compared to other protein-peptide interactions in this study or compared to experimentally validated results from the literature?).
Response: Line 54-55: Dear reviewer, this comparison was based only on the in-silico approaches which was because of High number of H-bonds compared to other protein-peptide interactions. Furthermore, the abstract section has been revised for English proofreading on order to make the statements understandable for the readers.
Reviewer 3 Report
The authors attempt to predict potent vaccine candidates against BaeR by the application of bioinformatic tools and databases. The computational prediction and evaluation are very meaningful to vaccine design, though the authors acknowledge the limitation and need the support of further experimental validation. The authors present their approaches clearly and provide relatively sufficient discussions. However, there are some major concerns that need to be carefully addressed:
- The authors applied many computational tools, but the introductions of these tools are not enough, for example, how solid and reliable these tools are? Any limitations of these tools? Why did the authors pick them? Any achievements have been made by these tools in other vaccine-related studies? Some tools have no introduction, like SOLPro and BepiPred. Some tools' background knowledge is only mentioned in the discussion section, which affects the understanding of this paper.
- The authors claim it's a novel attempt with these computational tools. But the authors do not tell clearly where the novelty from. Which steps are the first attempt? Which steps are following the other literature? Is the novelty from the combination of these prediction and evaluation tools? Or is the novelty from the first design again BaeR protein to tackle and mitigate MDR strains of A. baumannii?
- The authors are not certain about whether IEDB B-cell is a useful tool to predict epitopes (Line 288). But this is a very critical step in the whole design. The reference [32] and [33] seem not strong supports here. If they are, could the authors explain how IEDB helps to predict epitopes in [32] and [33]?
- In section 3.1, the authors do not explain clearly why they picked up two epitopes, E1 and E2, from 38 epitopes. The complementary explanation In the discussion section (Line 285~288) should be put in section 3.1. The authors also mention they designated two epitopes based on their previous studies[30, 31], but they do not provide any explanations on those studies.
Minor:
- In Line 208-209, the authors mention the default threshold of 0.5. But the VaxiJen threshold in Table 2 is 0.4. Is it a typo? Could the author give more explanation about the threshold? How it is related to score? Is 0.4706 a high enough score in VaxjJen?
- In Line 292, AlgPred is for the prediction of allergenicity, not for the prediction of antigenicity.
- In Section 3.4, the authors should provide some discussion on the result that transmembrane signals were not detected
- In Figure 2, the authors should briefly tell the meaning of low C-score, S-core, and Y-score
Author Response
Reviewer 3
Comments and Suggestions for Authors
The authors attempt to predict potent vaccine candidates against BaeR by the application of bioinformatic tools and databases. The computational prediction and evaluation are very meaningful to vaccine design, though the authors acknowledge the limitation and need the support of further experimental validation. The authors present their approaches clearly and provide relatively sufficient discussions. However, there are some major concerns that need to be carefully addressed:
- The authors applied many computational tools, but the introductions of these tools are not enough, for example, how solid and reliable these tools are? Any limitations of these tools? Why did the authors pick them? Any achievements have been made by these tools in other vaccine-related studies? Some tools have no introduction, like SOLPro and BepiPred. Some tools' background knowledge is only mentioned in the discussion section, which affects the understanding of this paper.
Response: At line 142-144: SOLpro predicts the propensity of a protein to be soluble upon overexpression in E. coli using a two-stage SVM architecture based on multiple representations of the primary sequence. It predicts about the solubility level of selected epitopes.
Furthermore, at line 171-175, Following statement has been added in the revised manuscript. “Bepipred-Parameters such as hydrophilicity, flexibility, accessibility, turns, exposed surface, polarity and antigenic propensity of polypeptides chains have been correlated with the location of continuous epitopes. This has led to a search for empirical rules that would allow the position of continuous epitopes to be predicted from certain features of the protein sequence.”
- The authors claim it's a novel attempt with these computational tools. But the authors do not tell clearly where the novelty from. Which steps are the first attempt? Which steps are following the other literature? Is the novelty from the combination of these prediction and evaluation tools? Or is the novelty from the first design again BaeR protein to tackle and mitigate MDR strains of A. baumannii?
Response: Dear reviewer, As mentioned the novelty was about the first of its kind study to design epitope peptides from BaeR to tackle and mitigate MDR strains of A. baumannii. It is been added under the conclusion.
- The authors are not certain about whether IEDB B-cell is a useful tool to predict epitopes (Line 288). But this is a very critical step in the whole design. The reference [32] and [33] seem not strong supports here. If they are, could the authors explain how IEDB helps to predict epitopes in [32] and [33]?
Response: Line 318-320: Dear reviewer, thank you for highlighting the point. The applicability of the tool has been added under the discussion. Furthermore, the extra references have been removed from the revised version of manuscript.
- In section 3.1, the authors do not explain clearly why they picked up two epitopes, E1 and E2, from 38 epitopes. The complementary explanation In the discussion section (Line 285~288) should be put in section 3.1. The authors also mention they designated two epitopes based on their previous studies[30, 31], but they do not provide any explanations on those studies.
Response: Line 213-218: The section has been modified according to the suggestion.
Minor:
- In Line 208-209, the authors mention the default threshold of 0.5. But the VaxiJen threshold in Table 2 is 0.4. Is it a typo? Could the author give more explanation about the threshold? How it is related to score? Is 0.4706 a high enough score in VaxjJen?
Response: Line 235-237: Yes, it was a typo error. A sentence has been added in the revised manuscript.
- In Line 292, AlgPred is for the prediction of allergenicity, not for the prediction of antigenicity.
Response: Line 346-2347: The sentence has been corrected.
- In Section 3.4, the authors should provide some discussion on the result that transmembrane signals were not detected
Response: Line 264-271: More discussion has been added in the revised version of manuscript.
- In Figure 2, the authors should briefly tell the meaning of low C-score, S-core, and Y-score
Response: Line 269-271: A description has been added in the revised version of manuscript.
Round 2
Reviewer 2 Report
While the grammar and general language of the manuscript was greatly improved, the content of the manuscript showed minimal improvement since the previous version.
Even still, the manuscript lacks sufficient rationale for the computational tools and databases used throughout the study.
While the authors included examples of other studies that used some of the same bioinformatic tools, the manuscript still lacks any level of quantitative validation of their workflow.
Still, using the author’s descriptions within Materials and Methods and Results, the reader is unable to recreate the results shown in Table 1.
Still, the ‘Discussion’ fails to comment on the implications of their results. The authors should interpret their findings more deeply rather than simply stating what each of the webservers generated.
Author Response
1. While the grammar and general language of the manuscript was greatly improved, the content of the manuscript showed minimal improvement since the previous version.
Improvised as instructed.
2. Even still, the manuscript lacks sufficient rationale for the computational tools and databases used throughout the study.
More details added for the tools and databases.
3. While the authors included examples of other studies that used some of the same bioinformatic tools, the manuscript still lacks any level of quantitative validation of their workflow.
Qualitative validation done.
4. Still, using the author’s descriptions within Materials and Methods and Results, the reader is unable to recreate the results shown in Table 1.
Given under Results sections 3.2.1 and highlighted.

Reviewer 3 Report
The authors added descriptions and discussions to address my questions. The revision looks good to me.
The abstract has a grammar mistake (Line 59): "an alternative strategy to tackling and mitigating..." It should be "tackle" and "mitigate".
Author Response
Reviewer 3
Comments and Suggestions for Authors
The authors added descriptions and discussions to address my questions. The revision looks good to me. The abstract has a grammar mistake (Line 59): "an alternative strategy to tackling and mitigating..." It should be "tackle" and "mitigate".
Response: Dear reviewer, we would like to thank you once again for revising the updated version of our manuscript. We also would like to acknowledge that, after addressing the comments from you and other reviewers, the manuscript has been thoroughly improved.
We have corrected the mistakes at Line 57. The “tacking and mitigating” has been corrected as “tackle and mitigate”.